# Broadband multi-beam lens-assisted mmID enabling multi-gigabit backscatter data rates for next-generation wireless networks

Marvin Joshi ✉, Charles A. Lynch III, Kexin Hu, Yaw A. Mensah, John D. Cressler & Manos M. Tentzeris

The growth of next-generation Internet-of-Things (IoT) and digital-twin systems has created wireless environments where identification must sustain fiber-level data rates with low latency while operating at minimal energy cost and maintaining robust angular coverage. Conventional backscatter at microwave frequencies remains limited to megabit rates, and most millimeter-wave demonstrations operate with limited coverage. This work presents a lens-assisted millimeter-wave identification (mmID) system that unites multi-gigabit connectivity with wide solid-angle coverage. The design integrates a cross-polarized broadband antenna array with a dielectric lens, enabling multi-beam operation with angle-dependent modulation across $\pm 55°$ and a peak differential radar cross section of -13.4 dBsm. Demonstrated backscatter performance includes 4 Gbps 32-QAM at 5 m with an energy cost of 0.08 pJ bit$^{-1}$ and 1 Gbps operation over 20 m. Link-budget analysis projects 1 Gbps backscatter ranges up to 2.6 km under the 75 dBm EIRP permitted in 5G millimeter-wave systems, establishing an energy-efficient pathway for high-capacity, long-range wireless identification.

The growth of Internet of Things (IoT) infrastructures, together with the deployment of 5G networks, is expected to interconnect tens of billions of devices in the coming decade[1]. Projections estimate nearly 43 billion IoT connections by 2030, with cellular IoT alone exceeding 7 billion[2]. These large-scale networks form the basis for city-scale digital twins, enabling distributed sensing for environmental monitoring, traffic coordination, and asset tracking at fine resolution[3–5]. A central challenge in such systems is achieving communication that is simultaneously high-throughput and energy-efficient, as conventional approaches struggle to scale with increasing device density and data demands. Wired solutions such as fiber or Ethernet provide gigabit data rates but lack scalability and flexibility for dense, distributed deployments. Running cables to tens of billions of nodes across cities, factories, and logistics hubs incurs prohibitive installation and maintenance costs while restricting device mobility. Active radios incur significant energy overhead due to oscillators and mixers, typically consuming hundreds of picojoules to nanojoules per bit, which is incompatible with sustainable, large-scale IoT.

Shifting to backscatter communication substantially reduces energy consumption by eliminating active RF components, but existing implementations at sub-GHz and microwave frequencies remain constrained to kilobit–megabit data rates due to limited bandwidth and low-order modulation schemes[6–11]. These constraints limit the applicability of conventional backscatter for emerging IoT and digital-twin platforms that demand both high data throughput and scalable deployment. Millimeter-wave (mmWave) frequencies offer a promising pathway to overcome these limitations. Large spectral allocations in the 24–30 GHz 5G bands enable gigabit-level data rates, while high equivalent isotropic radiated power (EIRP), up to 75 dBm, allows concentrated energy delivery over extended ranges[12,13]. The growing deployment of mmWave base stations in commercial 5G networks highlights the availability of this infrastructure for backscatter-based

School of Electrical and Computer Engineering, Georgia Institute of Technology, Atlanta, GA 30332, USA. ✉e-mail: mjoshi5@gatech.edu

systems. Short wavelengths at mmWave frequencies also support compact, high-gain antennas suitable for small-form-factor tags, with beamforming and steering enabling spatial selectivity. Prior demonstrations have shown that mmWave backscatter can achieve multi-gigabit throughput with extremely low energy per bit[14,15]. However, these systems are typically constrained to highly directional operation, limiting connectivity across wide angular ranges.

To improve angular coverage and effective gain, lens-based and retro-directive architectures have been explored. Rotman lenses and Van Atta arrays reshape incoming wavefronts to enhance range and angular performance, enabling wide-angle backscatter with increased realized gain[16–22]. Three-dimensional mmWave lenses further extend this concept by providing coverage across both azimuth and elevation planes[23–28]. Despite these advances, reported systems remain largely limited to sub-gigabit data rates, as their antennas and semi-passive front-ends lack the bandwidth required to exploit the multi-gigahertz spectrum available at mmWave frequencies. As a result, existing approaches do not simultaneously achieve gigabit-level throughput, wide angular acceptance, and ultralow energy operation.

In this work, a broadband millimeter-wave identification (mmID) system is presented that combines multi-gigabit backscatter communication with wide solid-angle coverage at ultralow energy per bit. The proposed architecture, as seen in Fig. 1, integrates a cross-polarized broadband antenna array with a dielectric lens to enable multi-beam operation and angle-dependent modulation while maintaining a compact and scalable form factor. By unifying broadband mmWave backscatter with lens-enabled angular robustness, this system addresses the fundamental trade-offs between data rate, coverage, and energy efficiency that have limited prior backscatter platforms, establishing a practical foundation for high-capacity identification in future IoT, digital-twin, and Industry 4.0 environments.

## Results

### Broadband cross-polarized antenna and modulator front-end

At the core of the mmID is a broadband, cross-polarized antenna designed to operate across the full 26–30 GHz band. A broadband element is essential to sustain gigabit-level backscatter, since narrowband operation would constrain throughput and increase distortion under high-order modulation. Cross-polarization is critical at mmWave, as a co-polarized backscatter would be masked by strong Tx-Rx coupling from the reader. Using orthogonal feeds enforces polarization separation between the incident and re-radiated waves, thereby suppressing interference and enhancing the effective signal-

to-noise ratio (SNR)[18,29]. Thus, to meet these requirements, a single-layer capacitively-coupled patch antenna was implemented, as shown in Fig. 2(a). Capacitive edge coupling excites multiple resonances that broaden the impedance bandwidth, while the matching stub in the feed network compensates the mismatch of the coupled feed, improving return loss and yielding a flatter impedance profile across the operating band. Two orthogonal ports provide cross polarizations, ensuring efficient capture and re-radiation regardless of the polarization of the incident wave. The antenna was designed in CST Microwave Studio and fabricated on Rogers 3003($\varepsilon_r = 3.00$, $\tan \delta = 0.0013$), with thickness of 0.254 mm. The measured and simulated results are shown in Fig. 2(b), confirming broadband impedance matching and cross-polarized performance, with $S_{11}$ remaining below -10 dB across the 26–30 GHz range.

To generate the subcarrier required for load modulation, the antenna is interfaced with a compact FET-based modulation network. By toggling the antenna impedance with a low-frequency bias, the circuit shifts the backscattered signal away from the carrier, suppressing direct leakage and enabling recovery of high-rate modulated backscatter. The backscatter modulator comprises a GaAs pHEMT CE3520K3 from California Eastern Laboratories integrated with radial stubs and a parallel stub connection, dimensioned to switch the electrical length between $\lambda/4$ and $\lambda/2$, as shown in Fig. 2(c)[30]. In the unbiased state with $V_{GD}$ equal to zero, the conducting channel sets the stub to act as a $\lambda/4$ open at the junction, which enables transmission and suppresses reflection. When $V_{GD}$ is driven negative toward threshold, the channel is cut off and the stub instead behaves as a $\lambda/2$ short at the junction, which redirects the incident wave back toward the port. Additionally, $\lambda/4$ stubs are placed at the source terminals to act as radial-stub-based RF chokes, which isolate the RF path while still enabling baseband biasing between the gate and drain of the FET. Through this combination of wireless mixing and RF choking, the circuit provides the necessary amplitude contrast for subcarrier generation while keeping the layout compact and ultralow in power consumption[17,23,29].

The backscatter modulator was designed in Keysight Advanced Design System and fabricated on the same Rogers 3003 substrate as the broadband antenna. The measured and simulated results are shown in Fig. 2(d), where the reflection response is plotted as $V_{GD}$ is swept from 0 V to $-0.75$ V in $-0.25$ V steps. At zero bias, the device resides in its strong-reflection state with $S_{21}$ near $-20$ dB. As $V_{GD}$ is driven more negative, the device transitions into its weak-reflection state with $S_{21}$ improving to about $-2$ dB. The corresponding

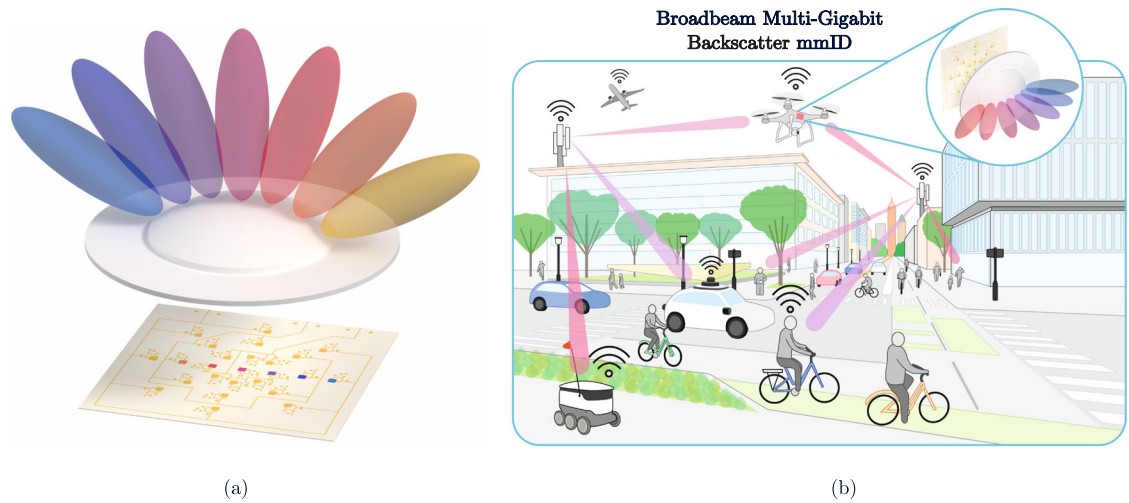

(a)                    (b)

**Fig. 1 | Lens-enabled mmID concept illustrating wide angular coverage and smart-city deployment. a** Diagram of the azimuth axis angular coverage Enabled by the Lens-Enabled mmID. **b** Diagram of the `smart' city infrastructure with dense deployment of the proposed mmID capable of gigabit backscatter data rates.

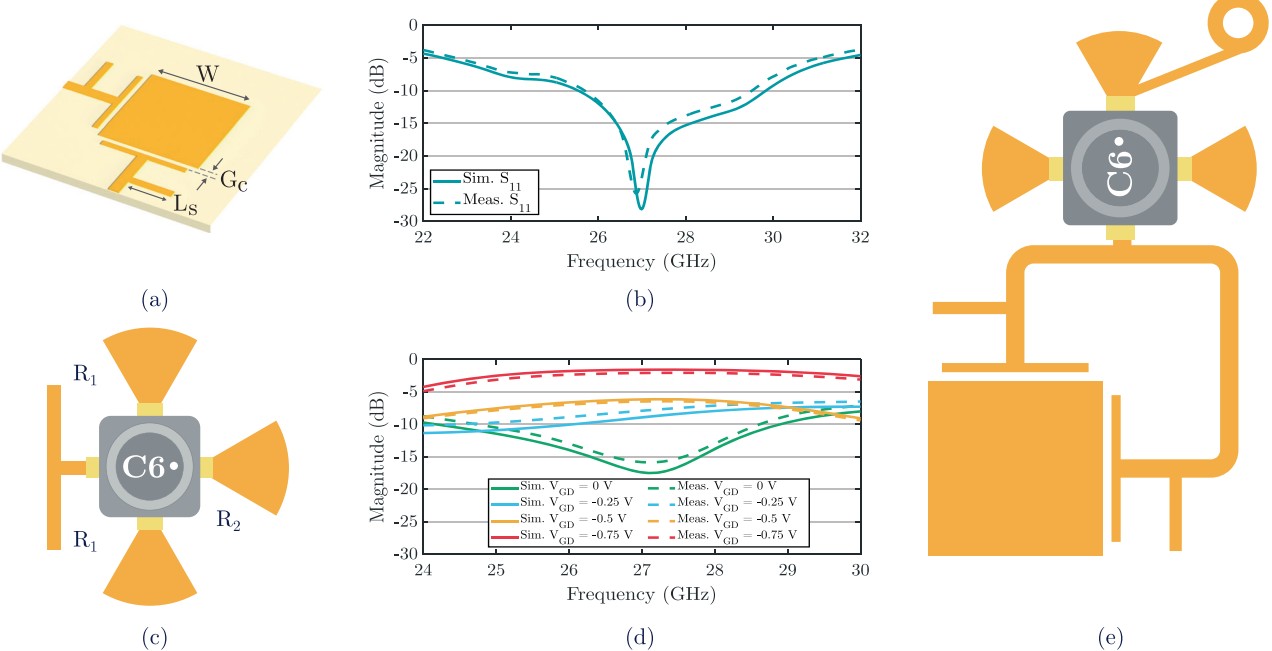

**Fig. 2 | Architecture and measured broadband response of the mmID pixel element. a** Layout of the cross-polarized capacitive-coupled patch antenna with dimensions $W = 2.85$ mm, $L_S = 1.1$ mm, and $G_C = 0.12$ mm. **b** Measured vs. Simulated $S_{11}$ results of the broadband antenna. **c** Layout of the FET-based mmWave modulator with dimensions $R_1 = 1.11$ mm and $R_2 = 1.24$ mm. **d** Measured vs. Simulated $S_{21}$ results of the mmWave modulator network. **e** Layout of the pixel back-scatter element, comprised of the broadband antenna and FET-based wireless mixer.

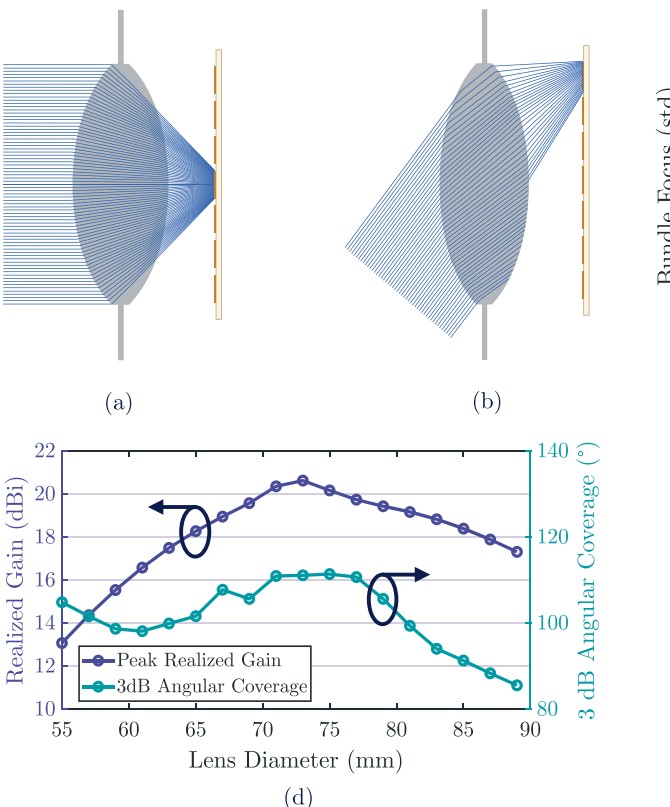

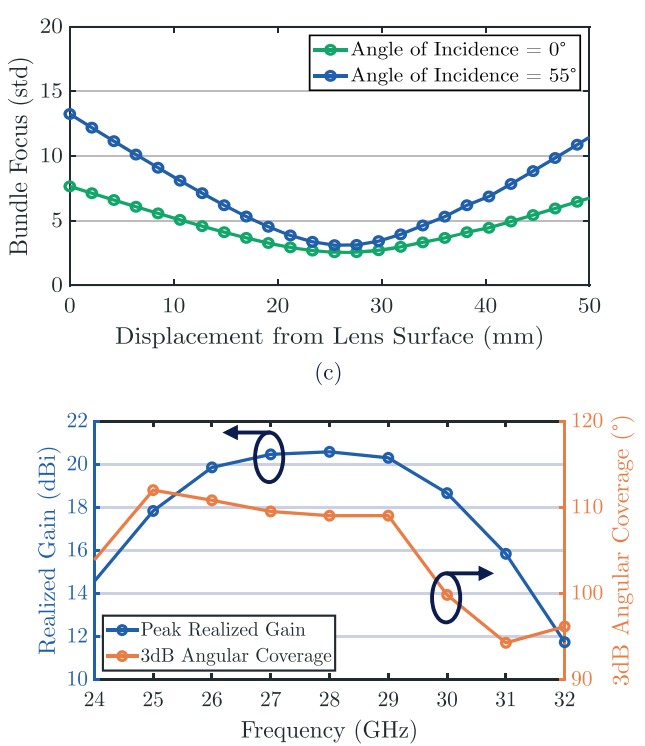

**Fig. 3 | Analysis of broadband focalization and angular response of the dielectric lens. a** Demonstration of the dielectric lens focalizing signals with angle of incidence = 0°. **b** Demonstration of the dielectric lens focalizing signals with angle of incidence = 55°. **c** Standard deviation of the ray bundle from the back surface of the dielectric lens at Angle of Incidence = 0° and 55°. **d** Simulated PTFE lens results of peak realized gain and 3 dBangular coverage results as a function of lens diameter at 27 GHz. **e** Simulated peak realized gain at boresight and 3 dB angular coverage as a function of frequency of the optimized PTFE lens.

modulation factor is given by

$$|\Delta\Gamma| = |\Gamma_{biased} - \Gamma_{unbiased}|, \tag{1}$$

where $\Gamma_{unbiased}$ corresponds to the zero-bias reflection state and $\Gamma_{biased}$ corresponds to the maximum applied bias, -0.75 V. Under these conditions, $|\Delta\Gamma|$ reaches a maximum of 0.71 at 27 GHz, with values above 0.65 maintained across the 26 GHz–29 GHz band. This ensures that the front-end bandwidth is not limited by the modulator. Instead, the overall system response is governed by the broadband antenna aperture, which later combines with the lens to enable multi-gigabit communication. When integrated with the broadband cross-polarized antenna, the FET-based backscatter modulator forms the complete pixel backscatter element, capable of enabling gigabit-level data rates, as demonstrated in Fig. 2(e). In the resulting pixel architecture, the antenna operates in a cross-polarized configuration because the two orthogonal feeding edges of the capacitive-coupled patch are tied together through a $\lambda/2$ microstrip transmission line, ensuring that the combined feed excites the patch in the orthogonal polarization of the incident wave. The FET-based modulation network is placed directly along this unified feed line so that the antenna and modulator function as a single coupled structure rather than as independent components. As a result, the applied bias sets whether the pixel resides in a strong-reflection state or a weak-reflection state. In the strong-reflection state, the pixel sends data by reradiating in the orthogonal polarization of the received wave, while in the weak-reflection state this cross-polarized reradiation is greatly reduced and no data is sent. This design provides the high-contrast binary backscatter response required for reliable multi-gigabit mmID operation.

## Dielectric lens-enabled mmwave focalization for high-rate backscatter systems

Gigabit backscatter at mmWave frequencies requires an antenna module that delivers both high gain and wide angular coverage. A dielectric lens provides an efficient solution, acting as a passive focusing element that concentrates incident energy onto the pixel[23,24,27,31–34]. Much like optical lenses that extend the field of view in imaging systems, the curved dielectric geometry focuses signals arriving from both boresight and oblique angles, thereby enhancing directivity with minimal loss compared to active beam-steering networks[35]. Material selection is critical since dielectric losses in backscatter systems penalize both the forward and return paths. Polytetrafluoroethylene (PTFE)($\varepsilon_r = 2.10$, $\tan\delta = 0.001$) was chosen as the lens material, since its exceptionally low loss enables focusing without degradation, which preserves long-range detectability.

The geometry of the lens is governed by its focal length, surface curvature, and the lateral dimension $W$, defined as the physical width of the planned mmID board. To ensure wide angular coverage while preserving a compact form factor, a target angular field of view (AFOV) of at least 110° was set. The angular field of view (AFOV) is given by

$$AFOV = 2\arctan\left(\frac{W}{2F}\right). \tag{2}$$

With $W$ set to 78 mm, the board length establishes the span of the device and directly sets the achievable AFOV. At this stage, $W$ provides the lateral constraint that ties the lens geometry to the system form factor. In the complete design, the mmID board will later integrate a pixel array to extend angular coverage and enable multi-beam operation, but for the lens synthesis it simply acts as the boundary condition linking board size to focal length. Based on this geometry, the focal length was chosen as 25 mm to realize the target AFOV, providing a balance between compact lens depth and wide angular range.

Ray-tracing simulations in Optometrika at 28 GHz were used to refine the lens geometry and confirm stable focusing across different angles. A sweep of the front and rear surface radii identified optimal values of 80 mm and -80 mm, as shown in Fig. 3(a) and (b). Here, ray trajectories are illustrated for the optimized lens for incident angles of 0° and 55°, both of which converge at the focal plane, confirming consistent focalization for on-axis and off-axis excitation. To further validate stability, a ray-bundle analysis was performed at 28 GHz. As illustrated in Fig. 3(c), the plot shows the bundle spread as a function of focal length, with the minimum occurring at the previously calculated value of 25 mm, confirming consistent focusing for both boresight and 55° incidence.

To further optimize the design for higher peak realized gain and wider 3 dB angular coverage, the optimized lens was modeled in CST Microwave Studio and a diameter sweep from 55 mm to 90 mm in 2 mm increments was performed. The results are shown in Fig. 3(d). Increasing the aperture size enhanced the realized gain, but the improvement diminished at larger diameters due to focal mismatch between the central and edge regions. A diameter of 74 mm provided the best balance, achieving up to 21 dBi of realized gain while preserving the 110° angular coverage. This size ensures efficient capture of both boresight and off-axis rays without enlarging the overall form factor. Broadband performance was then evaluated from 24 GHz to 32 GHz in 1 GHz steps, with the results plotted in Fig. 3(e). Sustaining gigabit-level backscatter requires stable behavior across wide bandwidths, since narrowband responses would limit throughput under higher-order modulation. The lens maintained realized gain above 20 dBi and angular coverage greater than 110° throughout 26 GHz to 29 GHz, confirming robust broadband performance. By combining compact geometry, optimized surface curvature, low-loss material properties, and broad operational bandwidth, the PTFE bi-convex lens achieves the gain and angular robustness required for gigabit-level backscatter operation.

## Pixel-array implementation for wide-angle backscatter

To extend the single pixel design into a practical mmID with wide angular coverage, a 25-element broadband cross-polarized pixel array was implemented, arranged in three concentric rings with a central element, as shown in Fig. 4(a). The rings have diameters of 26 mm, 52 mm, and 78 mm, each populated by eight evenly spaced elements. This geometry maximizes use of the aperture while preserving symmetry, which supports uniform angular coverage across both azimuth and elevation planes[23,24,27,31]. An inter-element spacing of 13 mm was selected as the minimum distance that suppresses coupling while still leaving room for bias routing on the top side of the board, avoiding extra layers or via transitions. Each ring of elements is biased together, with the center pixel grouped with the innermost ring, resulting in three independent bias groups. Three southwest connectors are mounted at the edge of the board and connected to the three groups to provide external bias access.

These groups map naturally to different angular sectors. Each pixel element exhibits a 3 dB beamwidth of about 18°, and when combined across the concentric rings and the focusing of the lens, the structure achieves continuous 3 dB angular coverage from boresight to ± 55°, corresponding to a solid angle of 2.68 sr. The physical dimensions of the optimized dielectric lens are shown in Fig. 4(b). By assigning separate bias signals to each group, the system enables angle-dependent modulation in which each sector of the array can operate with its own modulation state. This effectively creates spatial channels that can be selected electronically, without the complexity of mechanical steering or active beamforming. This design offers two key advantages. It allows multiple readers positioned at different angles to interrogate the tag without interfering with one another, since each angular sector can be assigned a distinct modulation state. It also provides flexibility to adapt the link: higher-order modulation can be applied along boresight where SNR is strongest, while simpler schemes are reserved for wider angles with reduced link margin. In both cases, the per-ring biasing ensures that modulation is only applied to

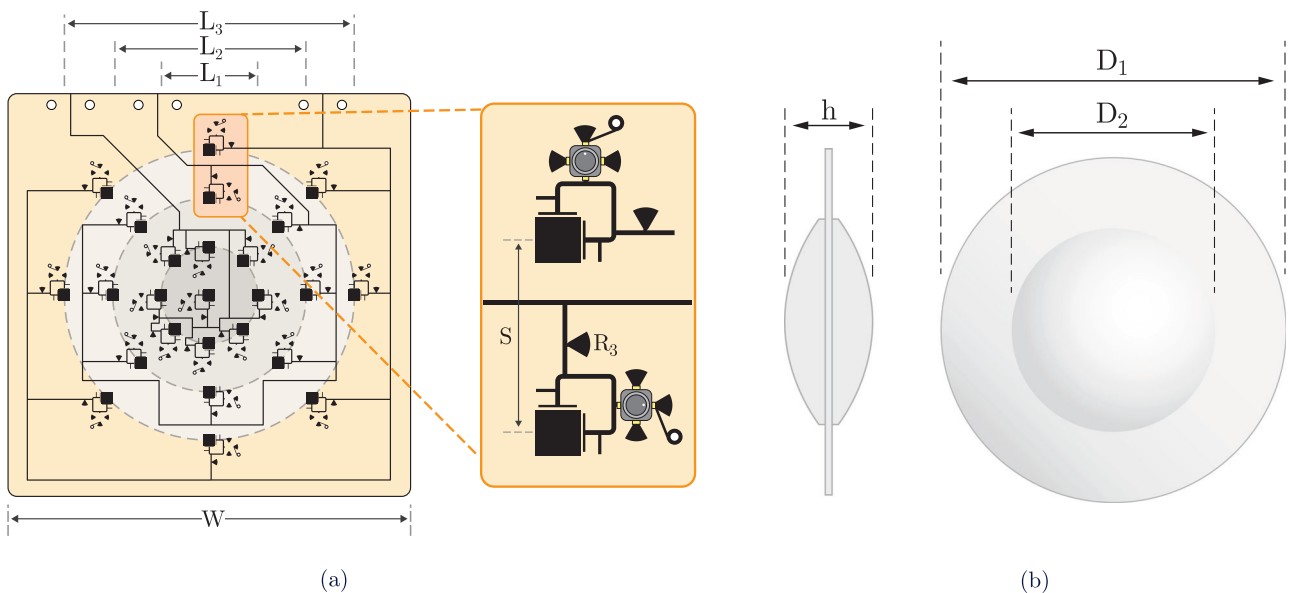

(a)

(b)

**Fig. 4 | Layout and geometry of the 25-element pixel array and PTFE dielectric lens. a** Proposed broadband cross-polarized mmID featuring 25 antenna elements with dimensions $L_1$ = 26 mm, $L_2$ = 52 mm, $L_3$ = 78 mm, $W$ = 90 mm, $S$ = 13 mm, and $R_3$ = 1.35 mm. **b** Proposed PTFE lens With dimensions labeled $D_1$ = 74 mm, $D_2$ = 120 mm, and $h$ = 25 mm.

illuminated sectors, improving efficiency and maintaining robust performance across the full angular range.

## Angular coverage characterization and differential RCS analysis of the lens-integrated mmID

To evaluate the radiation performance of the lens-integrated pixel array, a single row of the capacitive-coupled cross-polarized antennas was fabricated and positioned at the focal plane of the dielectric lens, located 25 mm from the aperture, as shown in Fig. 5(a). Each element in the row was connected directly to one port of the vector network analyzer in turn, while a 20 dBi A-INFO LB-180400-20-C-KF horn antenna driven from the other port served as the transmitter inside the anechoic chamber. The array-lens structure was rotated from − 70° to + 70° in 1° steps, and the angular response of each element was recorded across the sweep. At each rotation angle, the realized gain of the individual elements was measured at 26, 27, 28, and 29 GHz, with the corresponding radiation patterns shown in Fig. 5(b)-(e). The measurements exhibit strong agreement with simulations, confirming broadband stability across the band, with the lens-integrated array sustaining a 3 dB angular coverage of approximately ± 55° and achieving peak realized gains of 20.1 dBi, 20.7 dBi, 20.4 dBi, and 20.3 dBi at the four test frequencies.

After angular response characterization, the full mmID design was fabricated and tested for detectability, as shown in Fig. 6(a). The prototype was assembled with the dielectric lens and evaluated inside an anechoic chamber using two horn antennas: a 20 dBi A-INFO LB-180400-20-C-KF serving as the transmitter and an identical horn in cross-polarization serving as the receiver. The mmID was rotated from ± 70° in 1° steps. To assess detectability, the differential radar cross section (RCS) was extracted, capturing the contrast between the biased and unbiased states and directly reflecting the strength of the modulated backscatter relative to the static reflection. The differential RCS was then determined using

$$\Delta\sigma_{RCS} = \frac{\lambda^2 G_{mmID}{}^2 |\Gamma_A - \Gamma_B|^2}{4\pi}, \qquad (3)$$

where $\lambda$ is the free-space wavelength, $G_{mmID}$ is the realized gain of the mmID, and $\Gamma_A$ and $\Gamma_B$ represent the reflection coefficients in the two

states. A 12 inch diameter metal sphere served as a calibration reference to normalize the mmIds response. The measured and simulated differential RCS responses are plotted in Fig. 6(b), showing strong agreement across horizontal, vertical, and − 45° cuts. The prototype achieved a peak differential RCS of − 13.4 dBsm while maintaining a − 10 dB angular coverage of nearly ± 55°, corresponding to a solid angular coverage of 2.68 sr. To confirm stability across frequency, additional azimuth-plane measurements were performed at 26 to 29 GHz in 1 GHz steps. As shown in Fig. 6(c), simulation and measurement results remain consistent across the band, with the angular coverage and differential RCS preserved throughout. Together, the wide spherical angular coverage and broadband differential RCS highlight the effectiveness of the lens-integrated pixel array in delivering robust multi-beam interrogation to enable backscatter systems capable of sustaining gigabit data rates at long ranges.

## Backscatter link budget and read range analysis

With the angular coverage and differential RCS characterized, a link budget analysis was carried out to evaluate the achievable read range of the lens-integrated mmID. Two interrogation scenarios were considered: first, using a proof-of-concept (PoC) reader operating at an EIRP of 35 dBm, and second, utilizing the maximum allowed EIRP of 75 dBm in the 5G/mmWave bands. In both cases, the mmID was evaluated at boresight and at an incidence angle of 55°, corresponding to the angular limits of the measured coverage. These scenarios capture both the current hardware limits of the PoC reader, further explained below, and the upper bounds of regulatory-compliant transmission. Importantly, the analysis focuses on the maximum range at which a 1 Gbps backscatter link can be sustained. First, the incident power on the mmID as a function of distance is given by

$$P_{Rx,tag} = P_{Tx} + G_{Tx} + G_{mmID} + 10 n_{f_o} \log_{10}\left(\frac{\lambda_{f_o}}{4\pi R}\right) \qquad (4)$$

where $P_{Tx}$ is the transmitted power, $G_{Tx}$ is the transmit antenna gain, $G_{mmID}$ is the realized receive gain of the lens-integrated mmID, $n_{f_o}$ is the path loss exponent at the operating frequency $f_o$, $\lambda_{f_o}$ is the corresponding wavelength, and $R$ is the range between reader and tag. The resulting curves for both EIRP levels are plotted in Fig. 6(d),

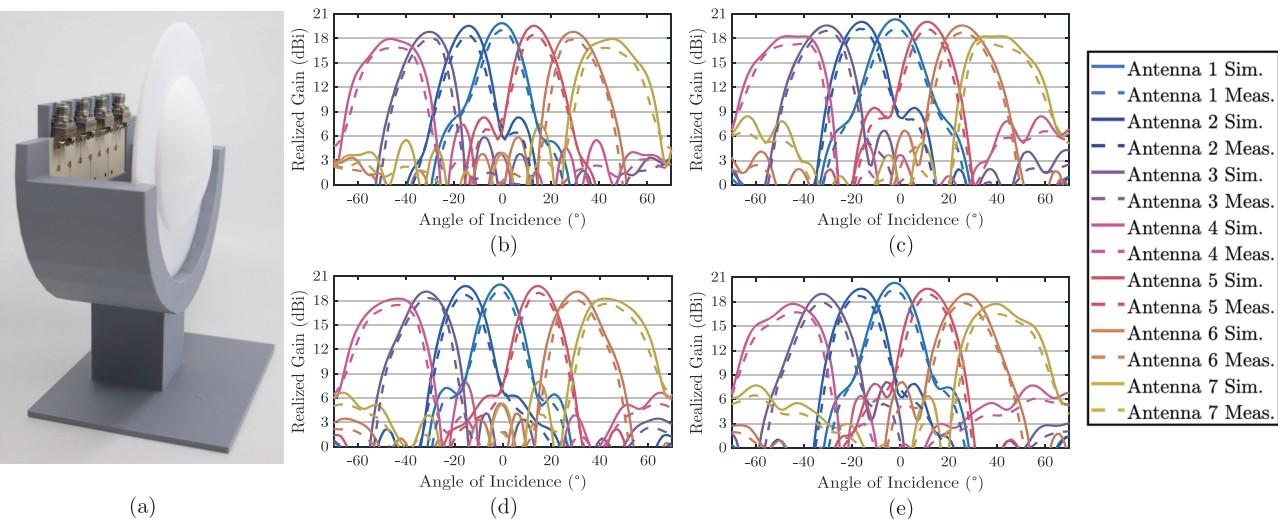

**Fig. 5 | Characterization of the angular gain of the lens-integrated pixel array. a** Fabricated capacitive-coupled pixel antenna array. Simulated vs. measured antenna gain across angle of incidence of the lens-based mmID at operational frequency of: **b** 26 GHz, **c** 27 GHz, **d** 28 GHz, and **e** 29 GHz.

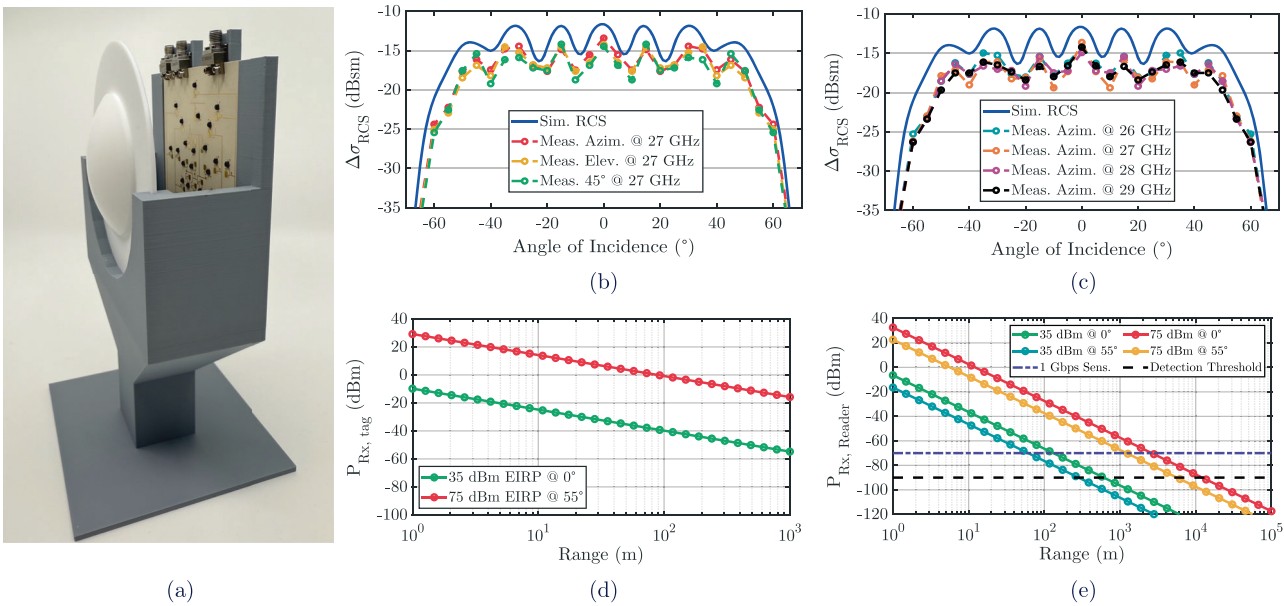

**Fig. 6 | Differential radar cross section and link-budget characterization of the mmID. a** Fabrication of the proof-of-concept gigabit per second data-rate backscatter mmID. **b** Simulated vs. measured differential RCS of the lens-enabled mmID as a function of angle of incidence for azimuth, elevation and 45° angular cuts. **c** Simulated vs. measured differential RCS of the lens-enabled mmID as a function of angle of incidence from 26--29 GHz. **d** Received power impinging onto the lens-based mmID with EIRP of 35 dBm and 75 dBm. **e** Received power vs. range from the lens-based mmID onto the PoC reader with EIRP of 35 dBm and 75 dBm.

illustrating the lens-enhanced impinging power onto the mmID as a function of distance.

The backscattered return was then calculated using the differential RCS previously characterized yielding

$$P_{Rx,Reader} = G_{Rx} + \Delta\sigma_{RCS}(\theta,\phi) + 10n_{f_o}\log_{10}\left(\frac{\lambda_{f_o}}{4\pi R}\right) + G_{LNA} \quad (5)$$

where $G_{Rx}$ is the receive antenna gain, $\Delta\sigma_{RCS}$ is the differential RCS as a function of azimuth $\theta$ and elevation $\phi$, and $G_{LNA}$ is the gain of the low-noise amplifier. This expression accounts for both the forward and return propagation paths and directly links the measured detectability of the mmID to the received power at the reader. As the mmID re-radiates the incident signal through the same lens-integrated aperture that receives it, the forward-path coupling is already incorporated into the differential RCS term described earlier. As a result, the formulation

does not include an additional $P_{Rx,tag}$ term, since the differential RCS already encompasses the complete receive-reradiate coupling of the mmID and naturally accounts for the forward-path interaction within $\Delta\sigma_{RCS}$. The reader's detection threshold was determined by the limits of the spectrum analyzer, with the resolution bandwidth RBW determining the effective noise bandwidth of the measurement. Accordingly, the minimum detectable signal at the analyzer input can be expressed as

$$S_{Rx,det} = -174 + G_{LNA} + NF + 10\log_{10}(RBW) + L_{Rx} + 20\log_{10}(EVM) \quad (6)$$

where $G_{LNA}$ is the gain of the LNA, $NF$ is the noise figure of the receive chain, $RBW$ is the analyzer resolution bandwidth in Hz, $L_{Rx}$ represents cable losses, and the last term represents the minimum SNR required to sustain low-BER 1 Gbps backscatter. Here, EVM denotes the error vector magnitude of the recovered constellation, which quantifies the

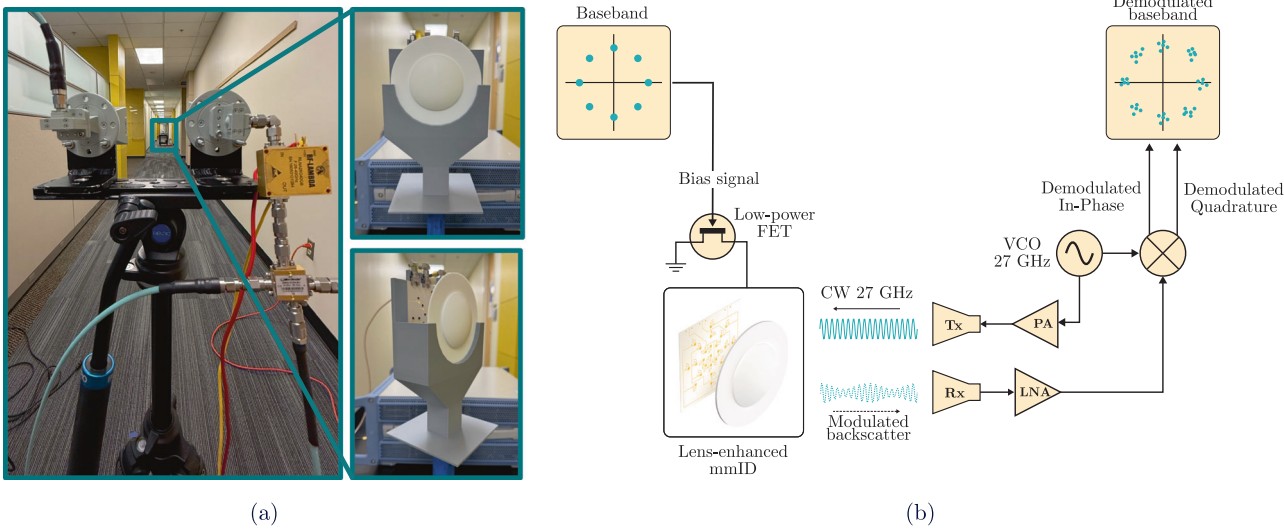

(a)

(b)

**Fig. 7 | Experimental setup and reader architecture for gigabit backscatter measurements. a** Experimental setup of the proposed lens-based mmID at incidence angles of 0° and 55° from the PoC reader. **b** Block diagram of the PoC reader transmitting and receiving chain to interrogate the lens-based mmID and demodulate the gigabit per second data-rate backscatter.

deviation of measured symbols from their ideal positions. Because this expression characterizes the detection threshold of the LNA-analyzer measurement chain, rather than the intrinsic sensitivity of a communication receiver, inclusion of RBW and LNA gain is appropriate and reflects the practical noise limitations of the measurement setup.

Since 32-QAM was the highest-order modulation experimentally demonstrated in this work, as will be presented in the upcoming section, it was chosen to determine the required SNR margin. In uncoded operation, BER values at or below $10^{-6}$ are generally regarded as sufficient for reliable demodulation, while in coded systems pre-forward error correction (FEC) thresholds as high as $10^{-4}$ to $10^{-5}$ are routinely adopted in practice[36,37]. For the measurement system, the noise floor was approximately -90 dBm, and achieving BER $< 10^{-6}$ with 32-QAM requires a minimum SNR of about 16.5 dB, corresponding to an effective detection threshold of -73.5 dBm.

The projected received power at the reader is shown in Fig. 6(e). Under the current PoC conditions at an EIRP of 35 dBm, the estimated maximum interrogation ranges were 100 m at boresight and 60 m at 55°. With the same detection threshold but an increased transmit power of 75 dBm EIRP, the ranges extend to 2.6 km at boresight and 1.2 km at 55°. These results confirm that the lens-enabled mmID supports both ultra-long-range detection and high-rate gigabit-level operation, while maintaining robust performance across wide angular coverage.

### Experimental setup and demonstration of angle-selective gigabit data rate backscatter

To evaluate the performance of the lens-integrated mmID under modulated backscatter operation, a CW testbed was assembled, as illustrated in Fig. 7. A Keysight 836640L Signal Generator produced a continuous-wave tone at 27 GHz, which was split by a PS2-53-450/15S power divider to provide a common local oscillator (LO) reference for both the transmit and receive chains. In the transmit path, the divided signal was amplified by a 16 dBm power amplifier and radiated through a 20 dBi A-INFO LB-180400-20-C-KF horn antenna. In parallel, a vector signal generator (VSG) provided the modulation bias waveforms to the three Southwest connectors of the mmID, driving each antenna group with a swing from 0 V to -0.75 V to produce modulated backscatter. The re-radiated signal was captured by a second 20 dBi horn antenna in cross-polarization, minimizing direct leakage from the transmitter and isolating the weaker backscattered response. The received waveform was amplified by a 43 dB RLNA26G40B low-noise amplifier,

downconverted using an ZMIQ-653H-E+ IQ mixer driven by the LO reference, and digitized on a Tektronix DPO70000SX oscilloscope for real-time constellation and spectrum analysis. A complete illustration of the modulation and demodulation sequence, including baseband formation, FET-based mixing, and baseband demodulation, is provided in Supplementary Note 1.

To benchmark the proposed system, three different modulation formats were applied to the three rings of pixel elements on the mmID: 32-QAM on the center and inner ring, 16-QAM on the middle ring, and 8-PSK on the outer ring. The signals were generated at a symbol rate of 800 Msym / sec, centered on a 2 GHz intermediate frequency (IF) subcarrier. Here, the IF denotes an offset carrier frequency used to shift the modulation away from the continuous-wave tone, simplifying separation of the backscattered signal from leakage at the reader and improving detection robustness. Each waveform was pulse-shaped with a square-root raised cosine (SRRC) filter with roll-off factor $\alpha = 0.25$, confining the occupied spectrum and minimizing inter-symbol interference for efficient high-rate operation[14,38]. The occupied bandwidth of the modulated waveform is given by

$$B_{occ} = (1 + \alpha) \cdot R_S \tag{7}$$

where $B_{occ}$ is the occupied bandwidth, $R_s$ is the symbol period, and $\alpha$ is the roll-off factor. For a symbol rate of $R_s = 800$ Msym / sec and $\alpha = 0.25$, the occupied bandwidth evaluates to $B_{occ} = 1$ GHz. The corresponding bit rates, $R_b$ are calculated as

$$R_b = R_s \cdot \log_2(M), \tag{8}$$

where $M$ denotes the number of constellation points, modulation order, in the chosen scheme. This results in 4.0 Gbps for 32-QAM, 3.2 Gbps for 16-QAM, and 2.4 Gbps for 8-PSK. A time-domain validation of the SRRC-shaped modulation chain is provided in Supplementary Note 2, confirming that the backscattered signal preserves the applied SRRC amplitude envelope.

The backscattered spectra and corresponding demodulated constellations for each modulation are shown in Fig. 8. The error vector magnitude (EVM), defined as the root-mean-square (RMS) difference between the measured constellation points and their ideal locations, was used to evaluate modulation accuracy, and the corresponding bit error rate (BER) was estimated from EVM using standard

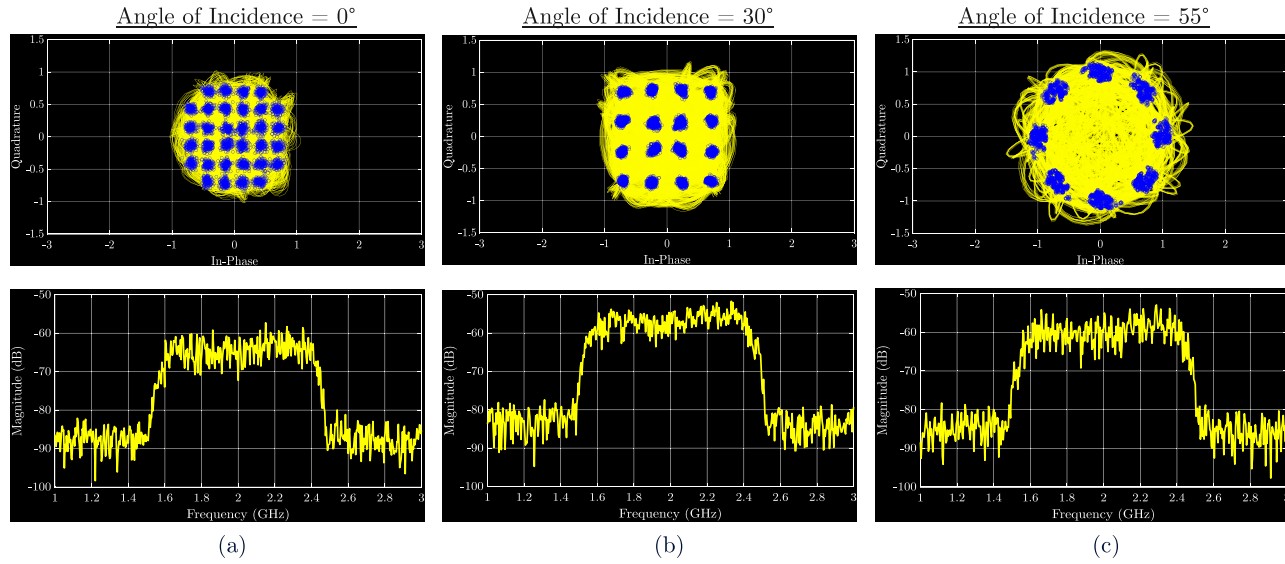

**Fig. 8 | Characterization of demodulated gigabit backscatter under angle-dependent biasing.** Measured demodulated gigabit backscatter constellation and spectra at: **a** 32-QAM constellation at angle of incidence = 0°, **b** 16-QAM at angle of incidence = 30°, **c** 8-PSK at angle of incidence = 55°.

analytical mappings[39]. For square M-QAM,

$$\text{BER}_{\text{QAM}} \approx \frac{4}{\log_2(M)}\left(1 - \frac{1}{\sqrt{M}}\right)Q\left(\sqrt{\frac{3\log_2(M)}{M-1}\cdot\frac{1}{\text{EVM}^2}}\right), \quad (9)$$

and for M-PSK,

$$\text{BER}_{\text{PSK}} \approx Q\left(\sqrt{\frac{2}{\text{EVM}^2}}\sin\frac{\pi}{M}\right). \quad (10)$$

For 32-QAM, the measured EVM was 7.1 %, corresponding to a BER of $2 \times 10^{-6}$. In the case of 16-QAM, the results showed an EVM of 6.3 % and a BER of $4.8 \times 10^{-13}$. With 8-PSK, the system exhibited an EVM of 5.6 %, giving a BER of $1.2 \times 10^{-8}$. Across all cases, low BER was achieved for reliable demodulation, demonstrating that the mmID achieves gigabit-level backscatter across its wide angular coverage. These results confirm that high-order modulations can be successfully recovered across the pixel array. In addition to demonstrating reliable gigabit demodulation, the system achieves ultra-low switching energy, quantified as the energy per bit. The total consumed power is expressed as

$$P_{\text{tot}} = I_{gs}V + \frac{1}{2}C_{gs}V^2 f_{\max}, \quad (11)$$

where $I_{gs}$ is the gate-source leakage current of the FET, $V$ is the applied bias voltage, $C_{gs}$ is the gate-source capacitance of the FET, and $f_{\max}$ is the maximum frequency component of the modulation. The corresponding energy per bit can then be expressed as

$$E_b = \frac{P_{\text{tot}}}{R_b}. \quad (12)$$

Although the modulation bias is distributed across all 25 pixel elements, only one illuminated element contributes to the backscattered signal at a given moment. The switching energy is therefore referenced to a single FET, since only one element is effectively active in the communication link at a time. The estimated energy per bit was 0.08 pJ/bit for 32-QAM operating at 4.0 Gbps, 0.10 pJ/bit for 16-QAM at

3.2 Gbps, and 0.13 pJ/bit for 8-PSK at 2.4 Gbps. These results demonstrate that the lens-integrated mmID achieves reliable wide-angle gigabit backscatter, with both low BER and ultralow energy per bit sustained across higher-order modulation formats.

### Demonstration of long-range gbps backscatter across boresight and off-axis incidence

The final characterization of the mmID focused on its ability to sustain gigabit backscatter communication over extended distances and wide angular coverage. In this measurement, the device was operated at a data rate of 1Gbps, driven by a 500 Msym / sec QPSK waveform shaped with the same SRRC filter with $\alpha = 0.25$ at an IF of 2 GHz. This signal was applied uniformly to all three southwest connectors of the mmID. The tag was placed at boresight and ranged from 2 m to 20 m in 2 m increments.

Figure 9(a) shows the received power as a function of distance, together with the theoretical fit for a two-way backscatter link. Path-loss exponents for indoor office environments are typically reported in the range of 1.4-2.0[16,21,40]. In the case of backscatter, the signal undergoes a two-way propagation, resulting in an effective exponent of approximately 3.2. The measured data follow this $R^{-3.2}$ trend closely, confirming the expected propagation behavior. For this measurement, while the system noise floor is -90 dBm, at least a 10.5 dB margin above the noise floor is required to maintain BER $< 10^{-6}$, giving an effective sensitivity of -79.5 dBm. Across the 2–20 m link, the received SNR remained above this threshold, ensuring reliable gigabit operation. The demodulated constellations at 2 m and 20 m are shown in Fig. 9(b) and (c), with EVM values of 1.3 % and 13.4 %, corresponding to BERs of below $10^{-18}$ for both. Even at 20 m, the signal remains resolvable, although with increased distortion compared to short ranges. To benchmark performance, the energy-per-bit was recalculated for this configuration and yielded 0.2 pJ/bit, which is higher than the earlier cases due to the increased modulation order, but remains very low for gigabit-level operation. The same measurement procedure was repeated at an incidence angle of 55°, representing the outer limit of the mmID's coverage. The received power versus distance is shown in Fig. 9(d), together with the same $R^{-3.2}$ fit, again showing good agreement with the measured data. At 2 m and 20 m, the demodulated constellations are shown in Fig. 9(e) and Fig. 9(f). The corresponding EVM values are 5.6 % and 27.2 %, which translate to BERs of $1.1 \times 10^{-19}$ and $1.02 \times 10^{-7}$, respectively. As expected, performance declines

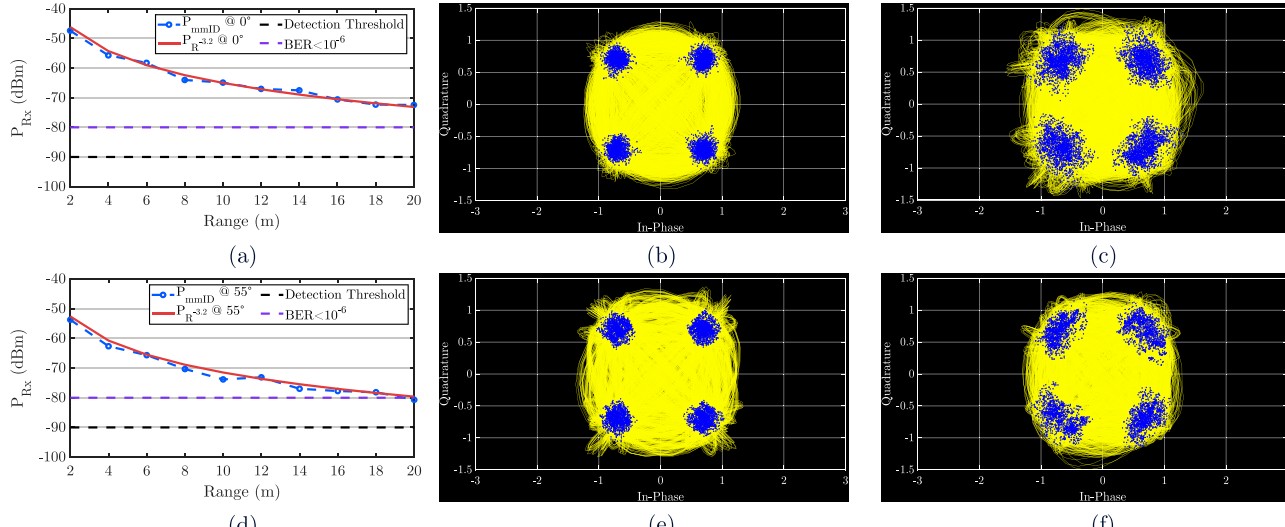

**Fig. 9 | Characterization of gigabit backscatter link performance as a function of range. a** Mean Measured Received Power of the Lens-Enabled mmID and its $R^{-3.2}$ Curve Fit with Respect to Range at Angle of Incidence = 0°. **b** Measured Demodulated Gigabit Backscatter QPSK Constellation at Range = 2 m and Angle of Incidence = 0°. **c** Measured Demodulated Gigabit Backscatter QPSK Constellation at Range = 20 m and Angle of Incidence = 0°. **d** Mean Measured Received Power of the Lens-Enabled mmID and its $R^{-3.2}$ Curve Fit with Respect to Range at Angle of Incidence = 55°. **e** Measured Demodulated Gigabit Backscatter QPSK Constellation at Range = 2 m and Angle of Incidence = 55°. **f** Measured Demodulated Gigabit Backscatter QPSK Constellation at Range = 20 m and Angle of Incidence = 55°.

**Table 1 | Comparison of state-of-the-art high-order, high data rate capable backscatter systems**

| Ref. | Frequency | Modulation | Data rate | Energy per bit | Range |
|---|---|---|---|---|---|
| 6 | 900 – 930 MHz | 16-QAM | 96 Mbps | 15.5 pJ bit⁻¹ | 1.24 m |
| 38 | 900 – 930MHz | 4-PAM | 1 kbps | 2.5 pJ bit⁻¹ | N/A |
| 41 | 900 – 930 MHz | 4-QAM | 400 kbps | 12.75 pJ bit⁻¹ | 4.5 m |
| 42 | 2.45 GHz | 16-QAM | 120 Mbps | 6.7 pJ bit⁻¹ | N/A |
| 15 | 24 GHz | ASK | 1 Gbps | 2400 pJ bit⁻¹ | 4.65 m |
| 14 | 24 – 28 GHz | 16-QAM | 2 Gbps | 0.17 pJ bit⁻¹ | 0.5 m |
| This work | 26–29 GHz | 32-QAM | 4 Gbps/1 Gbps | 0.08/0.2 pJ bit⁻¹ | 5 m/20 m |

compared to boresight, but the system still maintains 1 Gbps operation with BER below $10^{-6}$ throughout the 20 m link. These results confirm the robustness of the mmID across both boresight and wide-angle incidence, demonstrating gigabit backscatter operation consistent with theoretical two-way path-loss behavior.

Table 1 highlights that the proposed mmID outperforms prior high-data rate backscatter systems in maximum data rate, interrogation range and energy efficiency. Early sub-GHz and 2.45 GHz implementations[6,38,41,42] demonstrated data rates below 1 Gbps, with ranges typically under 5 m and energy costs on the order of several to tens of pJ bit⁻¹. More recent mmWave demonstrations at 24 GHz and 28 GHz[14,15] have reached data rates at or above 1 Gbps, but either incurred energy costs in the multi-nanojoule range or were limited to sub-5 m distances. In contrast, this work achieves sustained 1 Gbps backscatter over 20 m and extends to 4 Gbps over 5 m, while maintaining an energy consumoption of only 0.08 pJ bit⁻¹. These results make this the first mmWave backscatter platform to simultaneously combine gigabit-level throughput, long-range operation, ultra-low energy consumption per bit, and wide angular coverage.

## Discussion

In this work, the authors present a broadband, lens-enabled mmWave identification (mmID) platform that unifies multi-gigabit throughput with wide solid-angle coverage and record-low energy-per-bit consumption. The system integrates a pixelated array of cross-polarized capacitive-coupled patch antennas with a broadband dielectric lens to realize multi-beam operation and angle-dependent modulation.

Measurements confirm a peak differential RCS of − 13.4 dBsm across ± 55°, corresponding to a solid-angle coverage of 2.68 sr. The prototype achieved 4 Gbps using 32-QAM modulation at a distance of 5 m, and sustained 1 Gbps QPSK across 20 m at both boresight (0°) and off-axis (55°), with an energy cost of only 0.08 pJ bit⁻¹. Link budget analysis further projects 1 Gbps communication up to 100 m with the current reader, and up to 2.6 km under 75 dBm EIRP, underscoring the scalability of the approach. A key contributor to this long-range performance is the PTFE dielectric lens, which passively concentrates incident mmWave energy onto the pixel element in a manner analogous to an optical lens. The curved PTFE geometry maintains stable focusing for both boresight and oblique angles, and its very low dielectric loss preserves signal strength along both the forward and return paths. This passive gain improves the effective differential RCS without consuming power or adding RF complexity, enabling high-rate backscatter over extended distances while remaining fully compatible with ultra-low-power, semi-passive mmID operation.

By achieving fiber-level data rates, broad angular robustness, and pJ bit⁻¹ efficiency, the proposed mmID system sets a new benchmark for identification platforms. Conventional mmWave radios consume hundreds of mW in mixer stages at the device, whereas the proposed backscatter architecture eliminates mixers on the tag side, reducing front-end energy consumption by several orders of magnitude while still sustaining gigabit connectivity. This balance of efficiency, scalability, and wide-angle performance provides a favorable cost−benefit tradeoff and positions the technology as a practical solution for large-scale deployment. Looking ahead, several extensions can further

broaden the impact of this approach. Flat and metasurface-based lenses could miniaturize the optical front-end while preserving wide-angle performance. Angular multiplexing across multiple beams could enable simultaneous gigabit links to different readers, extending the concept beyond point-to-point operation. Integration with OFDM-style waveforms offers a direct pathway to compatibility with 5G and future 6G infrastructures. Finally, scaling to higher-order modulations such as 64-QAM or 128-QAM at extended ranges would push the throughput envelope while highlighting trade-offs between data rate and link reliability. Together, these directions position lens-enabled mmID as a key enabler of high-capacity, energy-efficient connectivity in smart-city networks, digital-twin infrastructures, and Industry 4.0 systems.

## Methods

### Design and simulation

All electromagnetic simulations were carried out using CST Microwave Studio and Keysight ADS. The cross-polarized capacitive-coupled patch antenna was modeled in CST using the time-domain solver, with open (add space) boundary conditions on all sides and discrete wave ports defined at the orthogonal microstrip feeds. The substrate was set to Rogers 3003 ($\varepsilon_r = 3.00$, $\tan \delta = 0.0013$) with a thickness of 0.254 mm, matching the fabricated design. Simulations were performed across the 24 GHz to 32 GHz band to extract S-parameters, radiation patterns, and polarization isolation.

For the lens analysis, a broadband dielectric structure was modeled in CST using PTFE ($\varepsilon_r = 2.10$, $\tan \delta = 0.001$). The lens geometry was set to the fabricated design, with a focal length of 25 mm. Plane-wave excitation was applied at boresight and at off-axis incidence up to 55°, while additional simulations were performed by exciting the patch antenna through its feed to confirm that the combined structure preserved retrodirective behavior. Parametric sweeps of lens diameter and curvature were carried out, and far-field monitors were used to evaluate angular gain shaping and coverage across the 24 GHz to 32 GHz band, ensuring retrodirective response across the full angular range. Circuit-level simulations of the FET-based modulator were carried out in Keysight ADS using the Modelithics nonlinear model for the CEL3520K3 transistor. Harmonic-balance simulations were performed with bias voltage swept from 0 V to $-0.75$ V to capture both the on and off states. S-parameters were extracted in each state, and matching stubs and transmission-line sections were tuned within ADS using the Modelithics parameters to maximize the difference in transmission response, thereby improving modulation depth and optimizing the mixing performance of the FET across the 24 GHz to 32 GHz band.

### Signal model of the continuous-wave backscatter testbed

The continuous-wave (CW) carrier generated by the reader is expressed as

$$x_{TX}(t) = A_{TX} \cos(2\pi f_c t + \theta_{TX}(t)), \tag{13}$$

where $x_{TX}(t)$ is the transmitted signal, $A_{TX}$ is the carrier amplitude, $f_c$ is the carrier frequency in Hz, and $\theta_{TX}(t)$ represents the phase noise of the source. At the mmID, the applied bias waveform drives the on-board FETs, resulting in the modulated backscatter

$$x_{BS}(t) = A_{BS} \, m(t) \cos\big(2\pi(f_c + f_{IF})t + \theta_{TX}(t) + \phi\big), \tag{14}$$

where $x_{BS}(t)$ is the reradiated backscatter signal, $A_{BS}$ is the back-scattered amplitude, $m(t)$ denotes the baseband modulation waveform applied through the bias line, $f_{IF}$ is the chosen intermediate frequency offset, $\theta_{TX}(t)$ is the transmit phase, and $\phi = -2\pi f_c R c^{-1}$ represents the one-way propagation phase delay over range $R$. After free-space propagation, the received signal at the reader can be expressed as

$$x_{RX}(t) = \frac{A_{BS}}{R^{2n}} \, m(t) \cos\big(2\pi(f_c + f_{IF})t + \theta_{TX}(t) + 2\phi\big), \tag{15}$$

where $R$ is the distance between reader and tag and $n$ is the path-loss exponent of the propagation channel. The total round-trip propagation phase is $2\phi = -4\pi f_c R c^{-1}$. Finally, IQ downconversion produces the complex baseband signal

$$x_{IQ}(t) = \frac{1}{2} A_{BS} m(t) \big[ \cos(\theta_{TX}(t) + \phi) + j \sin(\theta_{TX}(t) + 2\phi) \big], \tag{16}$$

where $x_{IQ}(t)$ contains the in-phase (I) and quadrature (Q) components of the modulated backscatter.

## Data availability

The data generated in this study are provided in the Supplementary Information/Source Data file. Any additional requests for information can be directed to, and will be fulfilled by, the corresponding author. Source data are provided with this paper.

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

## Acknowledgements

This work was supported by the National Science Foundation (NSF) through the Ideas Lab: ETAUS program under Grant No. 2322366 (M.M.T).

## Author contributions

M.J., C.A.L., and K.H. conceived the idea. M.J. designed the system, conducted the experiments and wrote the manuscript. M.J. and Y.A.M. built the wireless communication testbench for the experiments. C.A.L.

and K.H. analyzed the results. J.D.C. helped review the manuscript. M.M.T. supervised the research and contributed to the general concept and interpretation of the results. All authors reviewed the manuscript.

## Competing interests

The authors declare no competing interests.

## Additional information

**Peer review information** : *Nature Communications* thanks the anonymous reviewers for their contribution to the peer review of this work. A peer review file is available.

