## [Transparent Peer Review file · Nature Communications]

Broadband Multi-Beam Lens-Assisted mmID Enabling Multi-Gigabit Backscatter Data Rates for Next-Generation Wireless Networks

Corresponding Author: Mr Marvin Joshi

Version 0:

Reviewer comments:

Reviewer #1

(Remarks to the Author)

This work presents an interesting mmWave backscatter system, which achieves a data rate of 1 Gbps at a distance of 20m. This communication distance is much better than those in the reference [14,15]. It seems to me that the major difference is in the dielectric len. [14,15] used antenna array, and this work uses a dielectric len to focus the receive signal onto one antenn element. I am not sure why the dielectric len could bring such a significant improvement on the communication distance and energy efficiency. Could the authors provide a convincing explanation? How about a two-dimensional phased array is used, instead of the dielectric len.

Reviewer #2

(Remarks to the Author)

In this paper, a low power consumption and high data rate backscatter Tag is designed and measured. Here are my comments:

1. Overall, the paper lacks a detailed discussion of the design principles for key components, such as the patch antenna with FET. The authors should provide more details on the design of the patch antenna with FET.
2. The paper mentions using a dual-polarized antenna element. However, based on the description, I do not believe this element qualifies as "dual-polarized." A dual-polarized antenna in a communication system must have two independent feed ports, capable of simultaneously radiating and receiving two orthogonal polarization waves, with each polarization carrying independent or identical information. A key metric of dual-polarized antenna is polarization isolation. If the antenna element mentioned in this paper is truly dual-polarized, what is its polarization isolation metric? The authors should provide simulation or test results.
From the information provided, the element uses two orthogonal microstrip lines to excite the patch via edge coupling, generating two orthogonally polarized linear radiations. However, these two microstrip lines are connected together. Consequently, these two orthogonal linear polarizations will combine (based on their relative amplitude and phase) into a single linear polarization at an arbitrary angle, or even a circular polarization.
3. In Fig. 2(c) and Fig. 2(e), the orientation of the FET CE3520K3 is different. That is, the circuit connected to the FET's gate (pin 4, marked with a dot) is different in the two figures. The authors should check this. From the text, V_{GD} should be a negative value, but in the legend of Fig. 2(d), V_{GD} is positive. Please ensure the description is consistent.
4. The FET CE3520K3 is referred to as a "switch" in the paper, which I believe is inappropriate. Unlike a switch that only toggles between two or several discrete states, the FET in this paper is modulated by a continuous modulation signal, acting as a mixer. Therefore, calling it a "switch" is incorrect.
5. On page 3, it is mentioned that the substrate used in the antenna simulation is Rogers 4350B, whereas the substrate used in the FET simulation is Rogers 3003. Please explain why these two circuits were simulated using different substrate.
6. As shown in Fig. 6(d), when the Reader's transmitted EIRP is 35 dBm, the incident power at the Tag 10 meters away is approximately -20dBm. However, according to Equation (4), the free-space path loss at 27 GHz over 10 meters should be approximately 81 dB. This would imply an incident power at the Tag of approximately -46 dBm. Please explain this discrepancy.
7. In Equation (5), a term $P_{rx,tag}$ appears to be missing. Please check the equation and the corresponding calculations.
8. The calculation of receiver sensitivity in Equation (6) is incorrect. First, the thermal noise floor should be calculated based

on the received signal's bandwidth, not the spectrum analyzer's RBW (Resolution Bandwidth). Second, receiver sensitivity is determined by the receiver's Noise Figure (NF), signal bandwidth, and the required SNR, not by the limitations of the spectrum analyzer as mentioned. Moreover, the gain of the Reader's LNA should not be included in the sensitivity calculation. According to Equation (6), a higher LNA gain would incorrectly lead to a worse (higher) receiver sensitivity. Please review this equation carefully. By the same logic, the LNA gain should likely be excluded from Equation (5) as well. 9. In the experimental results of Fig. 8, it can be seen from Fig. 8(b) and Fig. 8(c) that the received signal power is greater than the received signal power in Fig. 8(a). Additionally, based on the received signal power, the SNR in Fig. 8(b) and Fig. 8(c) should be able to support higher-order modulation, yet low-order modulation was used. Please explain the reasons for this.

10. From Fig. 2(d), it can be seen that the relationship between the amplitude of the modulation signal and the amplitude of the reflected signal is not necessarily linear. For example, at 29 GHz (the center frequency of the reflected signal), the amplitude of the modulation voltage varies, but the amplitude of the reflected signal changes very little. This would cause a large discrepancy between the reflected signal's constellation diagram and an ideal one. In severe cases, calibration would be needed before use (refer to the constellation diagram in reference [6]). Please explain how, given the test results in Fig. 2(d), the received signal constellation diagram appears normal. Was some form of amplitude calibration technique used?

11. It is known that for a non-linear device like an FET, the non-linear effect will produce mixing when two signals are applied. Therefore, the Tag described in the paper will radiate not only the upper sideband (USB) signal centered at 29 GHz but also a lower sideband (LSB) signal centered at 25 GHz and other intermodulation signals. The paper does not include a spectrum measurement of the reflected signal before the demodulator. Please provide test results for the Tag's spurious emissions. If this LSB signal and other spurious signals exist, the application scenarios for this device will be severely limited, unless the authors can provide evidence that these emissions meet the requirements of regulatory authorities.

Version 1:

Reviewer comments:

Reviewer #1

(Remarks to the Author)
I have no more comment.

Reviewer #2

(Remarks to the Author)
The authors have already answered my questions, and the article has been revised accordingly. I have no further questions.

Authors' Response to Nature Communications Reviewers

REVIEWER COMMENTS

Reviewer #1 (Remarks to the Author):

This work presents an interesting mmWave backscatter system, which achieves a data rate of 1Gbps at a distance of 20m. This communication distance is much better than those in the reference [14,15]. It seems to me that the major difference is in the dielectric len. [14,15] used antenna array, and this work uses a dielectric len to focus the receive signal onto one antenn element. I am not sure why the dielectric len could bring such a significant improvement on the communication distance and energy efficiency. Could the authors provide a convincing explanation? How about a two-dimensional phased array is used, instead of the dielectric len.

The authors thank the reviewer for the comment. The extended communication distance demonstrated in this work is enabled in large part by the low-loss PTFE dielectric lens, which passively concentrates the incident mmWave energy onto the pixel element. As described in the manuscript, the curved PTFE geometry focuses signals arriving from both boresight and oblique angles, similar to an optical lens, and provides enhanced directivity with minimal insertion loss. The extremely low-loss PTFE material ($\epsilon_r = 2.10$, $\tan\delta = 0.001$) preserves signal strength across both the forward and return paths, and this directly strengthens the backscattered signal. However, the lens is only one part of the system. The long-range and high-throughput performance also relies on the broadband patch antenna and the high-speed FET-based modulator that are integrated within each

pixel. The broadband antenna provides efficient reception and reradiation across the operating band, and the FET modulator enables clean mixing that supports high-order constellations and high data rates. The combination of these three elements, namely the dielectric lens, the broadband antenna, and the high-speed modulator, enables the gigabit-class backscatter operation presented in this work.

The array-based tags in references [14] and [15] use smaller antenna structures with lower gain and narrower angular response, which limits the power delivered to the tag and reduces the effective reradiated cross-section. This results in lower capabilities in terms of maximum data rates and higher order modulation schemes. Because backscatter systems experience free-space attenuation on both the incoming and outgoing paths, even a moderate increase in passive gain leads to a significant improvement in achievable range.

Regarding the possibility of replacing the dielectric lens with a two-dimensional phased array, such an approach would require phase shifters, RF routing, and control circuitry, which introduce loss, increase cost, and consume power. One of the central goals of the proposed mmID is to remain semi-passive, ultra-low-power, and cost-efficient, making phased-array beamforming incompatible with the intended design constraints. The dielectric lens provides the required directivity and angular coverage with zero power consumption, and it works together with the broadband antenna and high-speed modulator to achieve the reported communication distance and efficiency.

Overall, the dielectric lens offers an efficient, low-loss, and power-free mechanism for enhancing both the incident signal on the mmID and the strength of the reradiated backscatter, which directly supports the extended communication distance and improved energy efficiency demonstrated in this work. This focusing enhancement works together with the broadband antenna element and the high-speed FET modulator, which provide effective reception, and enables the long-range and high-rate mmWave backscatter performance achieved in the measurements. An additional section has been added to the Discussion section of the paper and can also be found below:

In this work, the authors present a broadband, lens-enabled mmWave identification (mmID) platform that unifies multi-gigabit throughput with wide solid-angle coverage and record-low energy-per-bit consumption. The system integrates a pixelated array of cross-polarized capacitive-coupled patch antennas with a broadband dielectric lens to realize multi-beam operation and angle-dependent modulation. Measurements confirm a peak differential RCS of -13.4 dBsm across $\pm 55^\circ$, corresponding to a solid-angle coverage of 2.68 sr. The prototype achieved 4 Gbps using 32-QAM modulation at a distance of 5 m, and sustained 1 Gbps QPSK across 20 m at both boresight (0°) and off-axis (55°), with an energy cost of only 0.08 pJ/bit. Link budget analysis further projects 1 Gbps communication up to 100 m with the current reader, and up to 2.6 km under 75 dBm

EIRP, underscoring the scalability of the approach. A key contributor to this long-range performance is the PTFE dielectric lens, which passively concentrates incident mmWave energy onto the pixel element in a manner analogous to an optical lens. The curved PTFE geometry maintains stable focusing for both boresight and oblique angles, and its very low dielectric loss preserves signal strength along both the forward and return paths. This passive gain improves the effective differential RCS without consuming power or adding RF complexity, enabling high-rate backscatter over extended distances while remaining fully compatible with ultra-low-power, semi-passive mmID operation.

Reviewer #2 (Remarks to the Author):

In this paper, a low power consumption and high data rate backscatter Tag is designed and measured. Here are my comments:

1. Overall, the paper lacks a detailed discussion of the design principles for key components, such as the patch antenna with FET. The authors should provide more details on the design of the patch antenna with FET.

The authors thank the reviewer for the comment. More information on the integration of the FET-based modulator with the patch antenna has been added to the end of the section titled “Broadband Cross-Polarized Antenna and Modulator Front-End” and can also be found below:

“When integrated with the broadband cross-polarized antenna, the FET-based backscatter modulator forms the complete ‘pixel’ backscatter element, capable of enabling gigabit-level data rates, as demonstrated in Fig. 2(e). In the resulting pixel architecture, the antenna operates in a cross-polarized configuration because the two orthogonal feeding edges of the capacitive-coupled patch are tied together through a $\lambda/2$ microstrip transmission line, ensuring that the combined feed excites the patch in the orthogonal polarization of the incident wave. The FET-based modulation network is placed directly along this unified feed line so that the antenna and modulator function as a single coupled structure rather than as independent components. As a result, the applied bias sets whether the pixel resides in a strong-reflection state or a weak-reflection state. In the strong-reflection state, the pixel sends data by reradiating in the orthogonal polarization of the received wave, while in the weak-reflection state this cross-polarized reradiation is greatly reduced and no data is sent. This design provides the high-contrast binary backscatter response required for reliable multi-gigabit mmID operation.”

2. The paper mentions using a dual-polarized antenna element. However, based on the description, I do not believe this element qualifies as "dual-polarized." A dual-polarized antenna in a communication system must have two independent feed ports, capable of simultaneously radiating and receiving two orthogonal polarization waves, with each

polarization carrying independent or identical information. A key metric of dual-polarized antenna is polarization isolation. If the antenna element mentioned in this paper is truly dual-polarized, what is its polarization isolation metric? The authors should provide simulation or test results. From the information provided, the element uses two orthogonal microstrip lines to excite the patch via edge coupling, generating two orthogonally polarized linear radiations. However, these two microstrip lines are connected together. Consequently, these two orthogonal linear polarizations will combine (based on their relative amplitude and phase) into a single linear polarization at an arbitrary angle, or even a circular polarization.

The authors thank the reviewer for the comment and agree with the observation. The antenna elements are not used as true dual-polarized radiators in this design. Each antenna has two orthogonal feeding edges, but these edges are connected through the matching microstrip sections and operate together as a single path. Because of this, the structure does not provide two independent polarization channels. The manuscript has been updated to remove the term “dual-polarized” and refer to the antenna element simply as a cross-polarized patch used for the backscatter mmID. Additionally, the cross polarization is explained with more details in the response to Question 1.

3. In Fig. 2(c) and Fig. 2(e), the orientation of the FET CE3520K3 is different. That is, the circuit connected to the FET's gate (pin 4, marked with a dot) is different in the two figures. The authors should check this. From the text, V_{GD} should be a negative value, but in the legend of Fig. 2(d), V_{GD} is positive. Please ensure the description is consistent.

The authors thank the reviewer for the comment. The FET figure has been changed in Fig. 2(e) to its correct layout. Additionally, V_{GD} should be listed as a negative value. The caption for Fig. 2(d) has been changed to match this. The updated figure can be found below:

4. The FET CE3520K3 is referred to as a "switch" in the paper, which I believe is inappropriate. Unlike a switch that only toggles between two or several discrete states, the FET in this paper is modulated by a continuous modulation signal, acting as a mixer. Therefore, calling it a "switch" is incorrect.

The authors thank the reviewer for the comment and agree with the clarification. While the CE3520K3 behaves like a switch in our cross-polarized backscatter configuration, since the modulation either passes through or is suppressed depending on the FET bias, it is technically not an ideal RF switch. It is a FET whose gate-controlled impedance creates the mixing behavior observed in the measured results. To avoid any confusion, the manuscript has been revised to refer to the device simply as a FET based backscatter modulator, and the prior references of "switch" has been removed.

5. On page 3, it is mentioned that the substrate used in the antenna simulation is Rogers 4350B, whereas the substrate used in the FET simulation is Rogers 3003. Please explain why these two circuits were simulated using different substrate.

The authors thank the reviewer for the comment. A typo was made when listing the substrate used for the antenna. The correct substrate used for the entire mmID was Rogers 3003, and the antenna substrate has been updated in the text and can be found below:

"The antenna was designed in CST Microwave Studio and fabricated on Rogers 3003($\epsilon_r = 3.00$, $\tan \delta = 0.0013$), with thickness of 0.254 mm."

6. As shown in Fig. 6(d), when the Reader's transmitted EIRP is 35 dBm, the incident power at the Tag 10 meters away is approximately -20dBm. However, according to Equation (4), the free-space path loss at 27 GHz over 10 meters should be approximately 81 dB. This would imply an incident power at the Tag of approximately -46 dBm. Please explain this discrepancy.

The authors thank the reviewer for the comment. The forward-path incident-power curve in Fig. 6(d) was originally generated using the free-space expression in Equation (4), but the lens-enhanced receive gain of the mmID was unintentionally omitted. As the reviewer correctly noted, Equation (4) gives an isotropic received power of approximately -46 dBm at 10 meters, while the dielectric lens provides roughly 20 dBi of realized receive gain, resulting in the -25 dBm level shown in the figure. To address this, the forward-path equation has been updated to explicitly include the lens-integrated gain term, and the accompanying text has been revised to clarify that Fig. 6(d) plots the lens-enhanced incident power. The updated equation and revised paragraph can also be found below:

First, the incident power on the mmID as a function of distance is given by:

$$P_{R_{x,tag}} = P_{T_x} + G_{T_x} + G_{mmID} + 10n_{f_o} \log_{10} \left(\frac{\lambda_{f_o}}{4\pi R} \right)$$

where P_{T_x} is the transmitted power, G_{T_x} is the transmit antenna gain, G_{mmID} is the realized receive gain of the lens-integrated mmID, n_{f_o} is the path loss exponent at the operating frequency f_o , λ_{f_o} is the corresponding wavelength, and R is the range between reader and tag. The resulting curves for both EIRP levels are plotted in Fig. 6(c), illustrating the lens-enhanced impinging power onto the mmID as a function of distance.

7. In Equation (5), a term $P_{R_{x,tag}}$ appears to be missing. Please check the equation and the corresponding calculations.

The authors thank the reviewer for the comment. In a backscatter link, the receive power at the reader is governed by the *two-way* radar equation, where the forward-path power incident on the tag and the re-radiated power from the tag are combined through the radar cross section. For this reason, the intermediate term $P_{R_{x,tag}}$ does not explicitly appear in Equation (5). Instead, the forward-path power is already embedded within the differential radar cross section through Equation (3), repeated here for clarity:

$$\Delta\sigma_{RCS} = \frac{\lambda^2 G_{mmID}^2 |\Gamma_A - \Gamma_B|^2}{4\pi}$$

Because the mmID is a passive scatterer, the same physical aperture (lens + antenna) is used for receiving and re-radiating. As a result, the effective scattering strength is proportional to $P_{\text{Rx,tag}}$, which inherently incorporates:

- the receive-side gain of the lens-integrated mmID (forward-path enhancement),
- the modulation contrast introduced by the modulator network (through $|\Gamma_A - \Gamma_B|^2$)
- the re-radiation gain of the mmID (backward-path enhancement).

Thus, $\Delta\sigma_{\text{RCS}}$ already captures the complete effect of the power delivered to the pixel, and inserting an explicit term would double-count the forward-path contribution. To ensure clarity, the manuscript has been updated to explain that Equation (5) uses the differential RCS, which intrinsically contains both forward and return propagation effects, and can be found below:

The backscattered return was then calculated using the differential RCS previously characterized, yielding:

$$P_{\text{Rx,Reader}} = G_{\text{Rx}} + \Delta\sigma_{\text{RCS}}(\theta, \phi) + 10n_{f_o} \log_{10} \left(\frac{\lambda_{f_o}}{4\pi R} \right) + G_{\text{LNA}}$$

where G_{Rx} is the receive antenna gain, $\Delta\sigma_{\text{RCS}}$ is the differential RCS as a function of azimuth θ and elevation ϕ , and G_{LNA} is the gain of the low-noise amplifier. This expression accounts for both the forward and return propagation paths and directly links the measured detectability of the mmID to the received power at the reader. **As the mmID re-radiates the incident signal through the same lens-integrated aperture that receives it, the forward-path coupling is already incorporated into the differential RCS term described earlier. As a result, the formulation does not include an additional $P_{\text{Rx,tag}}$ term, since the differential RCS already encompasses the complete receive-radiate coupling of the mmID and naturally accounts for the forward-path interaction within $\Delta\sigma_{\text{RCS}}$.**

8. The calculation of receiver sensitivity in Equation (6) is incorrect. First, the thermal noise floor should be calculated based on the received signal's bandwidth, not the spectrum analyzer's RBW (Resolution Bandwidth). Second, receiver sensitivity is determined by the receiver's Noise Figure (NF), signal bandwidth, and the required SNR, not by the limitations of the spectrum analyzer as mentioned. Moreover, the gain of the Reader's LNA should not be included in the sensitivity calculation. According to Equation (6), a higher LNA gain would incorrectly lead to a worse (higher) receiver sensitivity. Please review this equation carefully. By the same logic, the LNA gain should likely be excluded from Equation (5) as well.

The authors thank the reviewer for the comment. The reviewer is correct that a true receiver sensitivity should be defined from the signal bandwidth, receiver noise figure, and required SNR, and is not determined by the spectrum analyzer's RBW or by the gain of the external LNA. To avoid confusion, the authors have clarified that Equation (6) does not represent the intrinsic sensitivity of a communication receiver. Instead, it describes the minimum detectable backscatter level of the measurement chain used in the experiments, which consists of the reader's LNA, cabling, and the spectrum analyzer operating with a fixed RBW. Because this expression is referenced to the analyzer input, the inclusion of RBW and LNA gain is appropriate for characterizing the measurement detection threshold, rather than a receiver sensitivity. The text has been updated to reflect this distinction, and the terminology in the manuscript has been revised accordingly, which can also be found below:

The reader's detection threshold was determined by the limits of the spectrum analyzer, with the resolution bandwidth (RBW) setting the effective noise bandwidth of the measurement. Accordingly, the minimum detectable signal at the analyzer input is given by:

$$S_{R_{x,det}} = -174 + G_{LNA} + NF + 10\log_{10}(RBW) + L_{Rx} + 20\log_{10}(EVM)$$

where G_{LNA} is the gain of the LNA, NF is the noise figure of the receive chain, RBW is the analyzer resolution bandwidth in hertz, L_{Rx} represents cable losses, and the last term reflects the minimum SNR required to sustain low-BER 1 Gbps backscatter. Here, EVM denotes the error vector magnitude of the recovered constellation, which quantifies the deviation of measured symbols from their ideal positions. Because this expression characterizes the detection threshold of the LNA-analyzer measurement chain rather than the intrinsic sensitivity of a communication receiver, inclusion of RBW and LNA gain is appropriate and reflects the practical noise limitations of the measurement setup.

Since 32-QAM was the highest-order modulation experimentally demonstrated in this work, as detailed in the following section, it was used to determine the required SNR margin. In uncoded operation, BER values at or below 10^{-6} are generally regarded as sufficient for reliable demodulation, while in coded systems pre-FEC BER thresholds as high as 10^{-4} to 10^{-5} are routinely adopted in practice. For the measurement system, the noise floor was approximately -90 dBm, and achieving $BER < 10^{-6}$ with 32-QAM requires a minimum SNR of about 16.5 dB, corresponding to an effective detection threshold of -73.5 dBm.

The projected received power at the reader is shown in Fig. 6(e). Under the current proof-of-concept conditions at an EIRP of 35 dBm, the estimated maximum interrogation ranges were 100 m at boresight and 60 m at 55° . With the same detection threshold but an

increased transmit power of 75 dBm EIRP, the ranges extend to 2.6 km at boresight and 1.2 km at 55°. These results confirm that the lens-enabled mmID supports both ultra-long-range detection and high-rate gigabit-level operation while maintaining robust performance across wide angular coverage.

9. In the experimental results of Fig. 8, it can be seen from Fig. 8(b) and Fig. 8(c) that the received signal power is greater than the received signal power in Fig. 8(a). Additionally, based on the received signal power, the SNR in Fig. 8(b) and Fig. 8(c) should be able to support higher-order modulation, yet low-order modulation was used. Please explain the reasons for this.

The authors thank the reviewer for the comment. The purpose of Fig. 8 was to demonstrate that different digital modulation formats can be applied to different rings of the mmID, rather than to show the maximum achievable modulation order. Although the received power in Figs. 8(b) and 8(c) is higher than in Fig. 8(a), the highest-order modulation that could be generated and processed in the measurement setup was limited by the capabilities of the waveform generator. The vector signal generator used in this work only supported a maximum symbol rate of 800 Msym/s, with 32-QAM being the highest-order constellation it could produce. As a result, lower-order modulations were used in Figs. 8(b) and 8(c) simply to illustrate the mmID's ability to accommodate different modulation formats across rings, not because of any limitation in the mmID hardware. The mmID itself can support higher-order modulation, but Fig. 8 was designed to highlight modulation flexibility rather than to maximize modulation order.

10. From Fig. 2(d), it can be seen that the relationship between the amplitude of the modulation signal and the amplitude of the reflected signal is not necessarily linear. For example, at 29 GHz (the center frequency of the reflected signal), the amplitude of the modulation voltage varies, but the amplitude of the reflected signal changes very little. This would cause a large discrepancy between the reflected signal's constellation diagram and an ideal one. In severe cases, calibration would be needed before use (refer to the constellation diagram in reference [6]). Please explain how, given the test results in Fig. 2(d), the received signal constellation diagram appears normal. Was some form of amplitude calibration technique used?

The authors thank the reviewer for the comment. We appreciate the concern regarding the nonlinear behavior shown in Fig. 2(d). In the proposed system, the tag does not perform analog RF amplitude modulation at the sideband frequencies. Instead, modulation is produced through reflection-based mixing of the 27 GHz continuous wave (CW) with a 2 GHz subcarrier applied to the FET gate. The pixel antenna is implemented in a cross-polarized configuration in which the FET is placed between the two orthogonal polarization paths of the capacitive-coupled patch; as a result, a change in the FET's reflection state determines whether energy is reradiated into the orthogonal polarization

that the reader receives. When the FET state is static, almost no energy couples into the receive polarization; when the FET state switches, the time-varying reflection coefficient generates the cross-polar component that carries the backscatter signal. The modulation depth is set by the reflection-coefficient contrast at the carrier frequency,

$$\Delta\Gamma = |\Gamma_1 - \Gamma_0|$$

and Fig. 2(d) shows that even moderate gate voltages such as -0.25 V provide sufficient contrast at 27 GHz to generate the strong mixing products observed at 25 GHz and 29 GHz. The small-signal behavior at 29 GHz therefore does not dictate constellation quality; the QAM waveform is recovered through coherent demodulation of the 2 GHz IF, consistent with the expected behavior of a nonlinear reflection mixer.

To further address the reviewer’s point, we have added time-domain measurements showing the applied and received waveforms, which can be found below. In these measurements, the tag is driven with a 32-QAM waveform shaped by a square-root raised-cosine filter, roll-off = 0.25, and upconverted to a 2 GHz subcarrier. The backscatter is measured at very low received power and therefore exhibits visible noise, however its envelope still follows the SRRC-shaped gate excitation, confirming that the reflection-coefficient variation at 27 GHz is sufficient, even at intermediate bias levels such as -0.25 V, to generate stable sidebands and undistorted constellations. These results have been added to the Supplementary Note 2 to directly illustrate the reflection-mixing process.

Supplementary Figure 3: Measured Time-domain signals used to verify reflection-based mixing: (a) Digitally generated 32-QAM symbol sequence from the vector signal generator (VSG). (b) SRRC-shaped 2 GHz subcarrier waveform applied to the FET gate with roll-off factor $\alpha = 0.25$. (c) Measured backscatter signal from the mmID demonstrating that its amplitude envelope follows the SRRC-shaped gate excitation.

11. It is known that for a non-linear device like an FET, the non-linear effect will produce mixing when two signals are applied. Therefore, the Tag described in the paper will radiate not only the upper sideband (USB) signal centered at 29 GHz but also a lower sideband (LSB) signal centered at 25 GHz and other intermodulation signals. The paper does not include a spectrum measurement of the reflected signal before the demodulator. Please provide test results for the Tag's spurious emissions. If this LSB signal and other spurious signals exist, the application scenarios for this device will be severely limited, unless the authors can provide evidence that these emissions meet the requirements of regulatory authorities.

The authors thank the reviewer for the comment. We agree that the nonlinear behavior of the FET modulator can generate both an upper and a lower sideband as well as higher order mixing products. In the original submission, we focused on the desired upper sideband used for demodulation and did not include a wideband spectrum measurement. To address this, we have performed an additional measurement using a 2-GHz ASK excitation and captured the reflected signal before any down-conversion. The resulting spectrum has been added to the supplementary information and can be found below.

From this measurement, the dominant component is clearly the upper sideband at 29 GHz. The corresponding lower sideband near 25 GHz is present but appears close to the noise floor. This is due to the lower sideband falls outside the high-gain region of this receive chain and therefore appears significantly attenuated. Additionally, all higher order mixing products and harmonics are also at or near noise floor levels. These observations indicate that unwanted emissions produced by the modulation process are strongly suppressed by the combined frequency response of the tag and the receive chain.

The current system is a laboratory proof-of-concept operating at modest power levels, and a full regulatory certification analysis is beyond the scope of this work. However, the measured spectrum demonstrates that spurious emissions are significantly attenuated, in addition to the lower sideband, and do not affect the operation of the mmID in the scenarios evaluated in this paper. An additional section has been added to the paper under Supplementary Note #1, including the backscatter signal modelling and the additional measurement showing the spectrum demonstrating suppressed lower sideband and dominant upper sideband.

Supplementary Figure 2: Measured RF Spectrum of the mmWave Backscatter Signal Showing the Dominant Upper Sideband and Suppressed Lower Sideband.